# Electrospun PCL-Based Vascular Grafts: In Vitro Tests

**DOI:** 10.3390/nano11030751

**Published:** 2021-03-16

**Authors:** Barbara Zavan, Chiara Gardin, Vincenzo Guarino, Tiberio Rocca, Iriczalli Cruz Maya, Federica Zanotti, Letizia Ferroni, Giulia Brunello, Juan-Carlos Chachques, Luigi Ambrosio, Vincenzo Gasbarro

**Affiliations:** 1GVM Care & Research, Maria Cecilia Hospital, 48033 Cotignola, Italy; chiara.gardin@unife.it (C.G.); ferronil@gmail.com (L.F.); 2Translational Medicine Department, University of Ferrara, 44123 Ferrara, Italy; zanottif@gmail.com; 3Institute of Polymers, Composites, and Biomaterials, National Research Council of Italy, Mostra d’Oltremare, Pad.20, V.le J.F.Kennedy 54, 80125 Naples, Italy; vguarino@unina.it (V.G.); cdiriczalli@gmail.com (I.C.M.); ambrosio@unina.it (L.A.); 4Division of Internal Medicine, St. Anna Hospital, 44123 Ferrara, Italy; rct@unife.it (T.R.); vincenzo.gasbarro@unife.it (V.G.); 5Department of Neurosciences, Dentistry Section, University of Padova, Via Giustiniani 2, 35128 Padova, Italy; giulia-bru@libero.it; 6Laboratory of Biosurgical Research (Alain Carpentier Foundation), Pompidu Hospital, University Paris Descartes, 75015 Paris, France; carloj@gmail.com; 7Department of Medical Sciences, Ferrara University, 44123 Ferrara, Italy

**Keywords:** Poly ε-caprolactone (PCL), electrospinning, vascular wall, in vitro validation

## Abstract

Background: Electrospun fibers have attracted a lot of attention from researchers due to their several characteristics, such as a very thin diameter, three-dimensional topography, large surface area, flexible surface, good mechanical characteristics, suitable for widespread applications. Indeed, electro-spinning offers many benefits, such as great surface-to-volume ratio, adjustable porosity, and the ability of imitating the tissue extra-cellular matrix. Methods: we processed Poly ε-caprolactone (PCL) via electrospinning for the production of bilayered tubular scaffolds for vascular tissue engineering application. Endothelial cells and fibroblasts were seeded into the two side of the scaffolds: endothelial cells onto the inner side composed of PCL/Gelatin fibers able to mimic the inner surface of the vessels, and fibroblasts onto the outer side only exposing PCL fibers. Extracellular matrix production and organization has been performed by means of classical immunofluorescence against collagen type I fibers, Scanning Electron-Microscopy (SEM) has been performed in order to evaluated ultrastructural morphology, gene expression by means gene expression has been performed to evaluate the phenotype of endothelial cells and fibroblasts. Results and conclusion: results confirmed that both cells population are able to conserve their phenotype colonizing the surface supporting the hypothesis that PCL scaffolds based on electrospun fibers should be a good candidate for vascular surgery.

## 1. Introduction

Currently, the treatment of patients suffering from aneurysmatic lesions of the aortic district is complex both from the point of view of the diagnostic framework and the therapeutic approach [1,2,3,4,5]. This involves the use of modular endoprostheses made of expanded non degradable polymers—i.e., polytetrafluoroethylene or polyethylene terephthalate—supported by a metal skeleton that is not easily positioned often, with long lead times and with the risk of possible future modular dislocations in the immediate or short to medium term [6,7,8,9]. Despite their relevant benefits in terms of chemical stability, long-term robustness, and non-toxicity, these devices frequently fail in vascular surgery due to the occurrence of uncontrolled biological events (i.e., late re-endothelialization, thrombosis phenomena, neointimal hyperplasia) [10]. From a clinical point of view, treated patients must undergo frequent controls to monitor the evolution of the aneurysmatic sac, which could be affected directly or through collateral circles, with the risk of a future rupture despite previous endovascular correction [11,12,13].

Ultimately, the current costs of these interventions are particularly burdensome both in prognostic terms for patients and in economic terms for the many resources that are employed during hospitalization, surgical/endovascular surgery, and follow-up.

Vascular tissue engineering has emerged as one of the most promising approaches to producing mechanically competent vascular substitutes [14,15,16]. Main attention on tissue engineering technology is on extracellular matrix that support cell growth, proliferation, differentiation, function. In our body, the extracellular matrix (ECM) is similar to the scaffold, and consists of 3D nanofibrous structure made of collagen and other biopolymers. Therefore, a 3D scaffold, if made of nanofibers, should provide a biomimetic structure resembling the ECM [17,18,19]. The nano-scale feature of a nanofibrous scaffold possesses high surface to volume ratio, which enhances cell adhesion and cell migration, and facilitates nutrient supply to the cells more efficiently [20]. The principal advantage gained by electrospun nanofibers is its biomimetic Extracellular Matrix structure. In order to improve its angiogenic potential it is possible modified its properties such as: Porosity with a porosity from 30 to 40 µm that is required to support the metabolite exchange; fiber orientation and heparin/gelatin functionalization to promote endothelial cells colonization; slower polymer degradation to support cell mobilization; scaffold stiffness to improve endothelial cells spreading and to support new sprout formation [21].

The use of slowly degradable polymers such as Poly ε-caprolactone (PCL) allows processing electrospun fibers with optimal morphological and biomechanical properties, in terms of flexibility and biodegradability, also able reproducing the fibrillary structure of the natural extracellular matrix of blood vessels [22]. In particular, PCL is a linear, hydrophobic, synthetic polymer that shows high mechanical strength. Although the electrospun PCL nanofibrous scaffold architecturally mimics the ECM in living tissues, its poor hydrophilicity caused a reduction in its ability of cell adhesion, migration, proliferation, and differentiation [23]. PCL is a Food and Drug Administration approved semi-crystalline hydrophobic polymer. It also has a low melting point and is biodegradable and bioabsorbable. This material has great potential as an implantable material in biomedical engineering. However, since the PCL is hydrophobic, it does not have good cellular activity. Thus, we carried out a surface modification process by using gelatin. Gelatin is a biocompatible, biodegradable, and natural biopolymer derived from collagen, a major component of native ECM. It is non-immunogenic and apparently retains informational signals such as the arginine–glycine–aspartic acid (RGD) sequence, which promotes cell adhesion, differentiation, and proliferation. Therefore, gelatin can be combined with PCL to obtain a composite scaffold having improved cell adhesion and proliferation properties, not significantly affecting functional response in terms of stability and flexibility [24,25,26]. Moreover, nanofibrous scaffolds show comparable mechanical properties respect to natural human blood vessels [27,28,29,30]. However, this study has not yet established biological assessments. For use in clinical applications, advanced surface modifications of bio-tubular scaffolds and pre-clinical experiments are necessary.

Starting from these assumptions, in the present work, we proposed the use of electrospinning to fabricate bioactive and biodegradable endoprosthesis able to exclude the aneurysmatic bag in the short and medium term, and to serve as a support (scaffolding) for the development of connective tissue intended to thicken the aneurysmatic wall preventing rupture in the long term.

## 2. Materials and Methods

### 2.1. Materials

Poly ε-caprolactone (PCL—Mn 45 kDa) and Gelatin Type B (~225 Bloom) from bovine skin in powder form, were all purchased from Sigma Aldrich (Milan, Italy), while organic solvents, 1,1,1,3,3,3-hexafluoro-2-propanol (HFIP) and chloroform (CHCl3) were supplied by J.T. Baker (Rodano, Italy). All products were used as received without further purification.

### 2.2. Electrospinning for Graft Fabrication

Electrospinning process has been preliminary optimized for the fabrication of fibrous membranes, made of PCL or PCL/Gelatin (50:50 w/w) with micro- and submicrometric diameters, respectively. Briefly, each polymer solution was placed in a 5 mL plastic syringe and forced to move through an 18 Gauge needle connected to a high voltage power supply. By the interaction between electrical forces and the polymer solution, fibers were formed, randomly collected over a grounded aluminum foil target until to obtain flat membranes. Process parameters used were summarized in Table 1. The process was carried out in a vertical configuration at 23–26 °C and 40/50% relative humidity degree. For the specimen fabrication, a commercially available electrospinning setup (Nanon01, MECC, Fukuoka, Japan) was equipped with an aluminum plate as flat collector.

Secondly, a rotating collector was properly selected for the fabrication of electrospun grafts with a specific geometry (diameter 6 mm; length 40 mm) suitable for clinical use. A tailored experimental setup was optimized to collect fibers directly onto a rotating collector by the sequential assembly of two different electrospun fibrous layers able to mimic peculiar structural features of the extracellular matrix of the vascular tissue. Hence, bilayered tubes were realized by collecting, in turn, PCL/Gelatin and PCL fibers onto a stainless-steel mandrel—i.e., diameter equal to 30 mm and rotating rate of 50 rpm—by a two sequential step deposition process to form the inner and outer layer of the final graft. Deposition times were properly defined for each step—2 h and 1 h, respectively—until they formed a graft wall 300 µm thick.

For preliminary biological essays, 6 mm discs were also cut from each layer and placed into 96-well tissue culture plates. All the specimens were sterilized in ethanol solution (70% *v*/*v*) for 30 min, washed three times with phosphate-buffered saline (PBS) and air dried before the use.

### 2.3. Cell Cultures

Fibroblast and Endothelials cells line were purchased from ATCC (Manassas, VA, USA) and seeded at a density of 4 × 10^5^ cells/piece on the membrane. Fibroblast on the side without gelatin and endothelial on the side treated with gelatin.

Fibroblast medium was formed by DMEM, supplemented with 10 vol % Fetal Bovine Serum (Bidachem-Spa, Milano, Italy) and 1% Penicillin/Streptomycin (P/S) (Sigma Aldrich, St. Louis, MI, USA).

Endothelial medium formed by DMEM containing 10% FBS plus 0.1 ng/mL human recombinant ECGF, 10 µg/mL human bFGF (Calbiochem, San Diego, CA, USA) and 100 µg/mL porcine heparin (Seromed, Berlin, Germany).

Cultures were incubated at 37 °C and 5% CO_2_ up to 7 days, media has been changed twice a week.

The control conditions are on tissue culture plates.

### 2.4. Immunofluorescence Staining

The samples were incubated in 2% bovine serum albumin (BSA, Sigma Aldrich, St. Louis, MI, USA) solution in PBS for 30 min at room temperature. The sections were then incubated with the primary antibodies in 2% BSA solution in a humidified chamber overnight at 4 °C. The following primary antibodies were used: mouse anti-CD31 antibody; rabbit monoclonal anti-vinculin antibody; mouse polyclonal anti-vonWillebrand factor antibody. Immunofluorescence staining was performed with secondary antibodies anti-mouse IgG DyLight 488 labeled and anti-rabbit IgG (H + L) rhodamine (TRITC)-conjugated in 2% BSA for 1 h at room temperature. Nuclear staining was performed with 2 μg/mL Hoechst H33342 (Sigma-Aldrich, St. Louis, MI, USA) solution for 2 min [31].

### 2.5. Morphological Analysis

To evaluate the morphological features of PCL and PCL/gelatin fibers, scanning electron microscopy (SEM) was used. Fiber morphology was qualitatively estimated by field emission scanning electron microscope (FESEM, QUANTA200, FEI, Eindhoven, The Netherlands) working at low voltage—lower than 10 kV—in order to prevent any sample alteration under the electron beam. Samples were dried in the fume hood for 24 h, mounted on metal stubs and sputter-coated with gold palladium and analyzed under high vacuum conditions using the secondary electron detector. Fiber diameter was measured from selected micrographs using Image analysis software (Image J, version 1.39). Fiber mean diameter were calculated from at least 30 measurements from three independent samples. For cell imaging, endothelial cells and fibroblast cultures were fixed in 2.5% glutaraldehyde in 0.1 M cacodylate buffer for 1 h, then progressively dehydrated in ethanol. Micrographs were obtained using a JSM JEOL 6490 SEM microscope (JEOL, Tokyo, Japan). The SEM analysis was performed at Centro di Analisi e Servizi Per la Certificazione (CEASC, University of Padova, Padova, Italy) [32].

### 2.6. Cell Shape Analysis

Cell shape was evaluated on samples stained for 40 min with 5 mg/mL phalloidin. Briefly, cells were fixed in 4% paraformaldehyde in PBS for 10 min, then permeabilized with 0.1% triton X-100 (Sigma-Aldrich, Saint Louis, MA, USA) in PBS for 30 min at room temperature. Phalloidin was then used for fluorescent staining of actin filaments, whilst nuclear staining was performed with 2 μg/mL Hoechst H33342 (Sigma-Aldrich) solution for 5 min. Images were acquired with the inverted optical microscope DMI4000 B (Leica Microsystems, Wetzlar, Germany). Then, ImageJ software was used to calculate cell area and different shape descriptors of at least 30 distinct cells. In detail, the Circularity (C), Roundness (R), and Solidity (S) of cells were calculated according to the following equations:Circularity = C = 4πA/P2(1)
where A is the cell area and P is the perimeter;
Roundness = R = a/b(2)
where a and b are the width and length of the minimum bounding, respectively;
Solidity = S = A/ConvexA(3)
where ConvexA is the area enclosed by the smallest shell that borders all the points of the cell [33].

### 2.7. Proliferation Assay

To determine the presence of viable cells in membrane samples the MTT (3-(4,5-dimetiltiazol-2-il)-2,5-difeniltetrazoli)-based proliferation assay was performed as described in Gardin et al. [31]. Briefly, samples were incubated for 3 h at 37 °C in 1 mL of 0.5 mg/mL MTT solution prepared in PBS. After removal of the MTT solution by pipette, 0.5 mL of 10% DMSO in isopropanol was added to extract the formazan in the samples for 30 min at 37 °C. For each sample, O.D. values at 570 nm were recorded in duplicate on 200 μL aliquots deposited in microwell plates using a multilabel plate reader (Victor 3, Perkin Elmer, Milano, Italy) [34].

### 2.8. Lactate Dehidrogenase (LDH) Activity

LDH activity was measured using a specific LDH Activity Assay Kit (Sigma Aldrich, St. Louis, MI, USA) at 3, 7, 14, 21, and 28 days of cell culture. All conditions were tested in duplicate. The culture medium was reserved to determine extracellular LDH activity. The intracellular LDH activity was estimated after cells lysis with the assay buffer contained in the kit. All samples were incubated with a supplied reaction mixture, resulting in a product where its absorbance was measured at 450 nm using Victor 3 plate reader [35].

### 2.9. ELISA Assays

The supernatants of the endothelial and fibroblast cultures were collected, centrifuged at 400× *g* for 10 min at 4 °C, then stored frozen for analysis of VEGF, production using commercial ELISA kits (Thermo Fisher Scientific, Waltham, MA, USA) according to the manufacturer’s instructions. Optical density (OD) values at 405 nm were measured using Victor 3 plate reader [36].

### 2.10. Beta-Galactosidase Staining (SA-b GAL) Staining

Beta-galactosidase staining was performed using the Senescence-associated β-galactosidase staining kit (Cell Signaling Technology, Danvers, MA, USA). DSPCs were fixed with a fixative solution, and stained with the β-Galactosidase staining solution at 37 °C over night. The number of positive and negative cells were then counted in five random fields under the microscope, and the percentage of SA-b GAL positive cells was calculated as the number of positive cells divided by the total number of cells counted [37].

### 2.11. ROS/RNS Assay

The OxiSelect™ In Vitro ROS/RNS Assay Kit (Cell Biolabs, Inc., San Diego, CA, USA) was used for quantifying total ROS/RNS free radical activity in the culture samples. The generation of ROS/RNS was measured using the dichlorodihydrofluorescin DiOxyQ (DCFH-DiOxyQ) probe following the manufacturer’s instructions. For the quantification of intracellular ROS/RNS, cells were lysed by sonication on ice, then centrifuged at 10,000× *g* for 5 min to remove insoluble particles, before incubation with the DCFH solution for 45 min at room temperature. For measuring the extracellular ROS/RNS production, cell culture supernatants were centrifuged at 10,000× *g* for 5 min, then incubated with the DCFH solution for 45 min at room temperature. During this incubation, the non-fluorescent DCHF is oxidized by ROS and/or RNS present in the samples, generating the fluorescent derivative 2′,7′-dichlorofluorescein (DCF), whose intensity is proportional to the concentration of total ROS/RNS levels within the samples. The fluorescence was recorded on a multilabel plate reader (Victor 3, Perkin Elmer, Milan, Italy) at 480 nm excitation/530 nm emission [38].

### 2.12. RNA Extraction and First-Strand cDNA Synthesis

Total RNA was isolated from cells grown onto scaffolds for 3 and 7 days using the total RNA purification Plus kit, and from hADSCs-monocytes before seeding onto bone granules (Norgen Biotek, Thorold, ON, Canada). The RNA quality and concentration of the samples were measured with the NanoDrop™ ND-1000 (Thermo Fisher Scientific, Munich, Germany). For each sample, 500 ng of total RNA was reverse-transcribed using an RT2 First Strand kit (Qiagen, Hilden, Germany) in a final reaction volume of 20 μL [39].

### 2.13. Real-Time PCR

Real-time PCR was performed according to the user manual of the Human Mitochondrial Energy Metabolism RT2 profiler PCR Array (Qiagen, Hilden, Germany) with a StepOnePlus™ Real-Time PCR System (Applied Biosystems™, Foster City, CA, USA) and using RT2 SYBR Green ROX FAST Master Mix (Qiagen). Thermal cycling and fluorescence detection were as follows: 95 °C for 10 min, followed by 40 cycles of 95 °C for 15 s, and 60 °C for 1 min. At the end of each run, a melting curve analysis was performed using the following program: 95 °C for 1 min, 65 °C for 2 min with optics off, 65 °C to 95 °C at 2 °C/min with optics on [39].

### 2.14. Statistical Analysis

All results are expressed as the mean ± standard deviation (SD) obtained from at least three independent experiments. Significant differences among groups were determined by analysis of variance (ANOVA), followed by post hoc Bonferroni tests. Student’s *t*-test was performed to determine the statistical significance between two samples. Different labels indicate * *p* < 0.05, ** *p* < 0.01, and *** *p* < 0.001.

## 3. Results

### 3.1. Scaffold Characterization

Figure 1 shows the fiber morphology of the inner and outer layer of vascular grafts fabricated via electrospinning. Both layers exhibited uniform and continuous spatial distribution of fibers with relevant differences in terms of fiber diameters. Quantitative measurements via image analysis reveal an evident difference in the fiber diameters from micrometric (5.11 ± 0.47) µm to sub-micrometric size scale (0.822 ± 0.11) µm, ascribable to the different dielectric permittivity of used solvents—i.e., chloroform or HFIP. In particular, higher permittivity of HFIP allows to bridge stronger interactions among polymer chains under the applied electrical forces, thus promoting a higher stretching of the polymer jet during the fiber deposition [40], and ultimately, a reduction of the final diameter of fibers—up to one order of magnitude, as confirmed by image analysis data.

Moreover, the combination of Gelatin macromolecules with PCL in solution to form bicomponent blended fibers allows drastically reducing instability phenomena of the polymer jet, typically induced by the use of highly permittive solvents [41], thus guaranteeing the formation of defect-less fibers with more homogeneous fiber diameter distributions, according to experimental evidences on similar bicomponent fibers [42,43].

### 3.2. Cell Shape

Endothelial cells and fibroblasts were isolated and seeded onto the two sides of the scaffolds in form of membrane: gelatinated side with smaller fibers with endothelial cells and non-gelatinated side with bigger fibers with fibroblasts. Morphology of the cells was evaluated at time 0 (Figure 2A for endothelial cells and Figure 2C for fibroblasts). As reported in Figure 2 endothelial cells showed a diameter size of approximately 20–30 nm, mostly homogenous inside the population, and fibroblasts a diameter size of about 10–30 nm. Positivity against CD31 (green staining) and vW (red staining), specific marker for mature endothelial cells (Figure 2B) confirmed the right commitment of the vascular cells, and positivity (green staining) for anti-fibroblastic markers showing the presence of a population of fibroblast well defined (Figure 2D).

A detailed analysis of cell shape was additionally performed (Figure 3). In particular, the Circularity, Roundness, and Solidity parameters were considered. As shown in Figure 2, when endothelial cells (white bars) were seeded onto the scaffolds and cultured up to 7 days, they assumed a short elongated morphology, as suggested by the Circularity and Roundness calculated values, by contrast fibroblasts assumed a more elongated morphology.

### 3.3. MTT

Metabolic activity was measured by means of MTT test after 3 and 7 days of culture (Figure 4). After 7 days of culture, a statistically significant increase in cell metabolic activity was recorded in the PCL scaffolds, in comparison to the activity at 3 days for both endothelial and fibroblast. No significative difference was found comparing the PCL scaffold condition to the normal classic plastic plate surface.

### 3.4. Intracellular and Extracellular LDH Activity

Damage potential of the scaffolds surface on cells was evaluated with LDH activity assay. Figure 5, related to intracellular LDH activity shows that endothelial cells and fibroblasts were able to produce metabolites if seeded onto both surfaces, with improved results after 7 days from seeding. In Figure 5, extracellular LDH activity is reported. Presence of LDH in extracellular compartment is generally correlated to a membrane damage, and this could be possibly due to the scaffold composition. The LDH activity measured in the culture medium confirms that there was no damage associated to the membrane.

### 3.5. SEM

SEM images showed how cells attached to the surface for the different samples. After 3 days of culture, cells showed a good distribution and large size. Both endothelial cells (Figure 6A–C) and fibroblasts (Figure 6C–E) were able to attach to the fibers with their filopodia, exhibiting a good ability to migrate and to secrete exosomes (black circles) onto their surfaces.

### 3.6. VEGF Production

In order to test the ability of endothelial cells to maintain their phenotype their VEGF secretion potential was analyzed by means its quantification in the cell culture medium up to 7 days by means of an ELISA test. As reported in Figure 7, VEGF concentration increased in the medium in a time-dependent manner, starting from a basal concentration of 180 ng/mL after 1 h and reaching the maximum level of 1800 ng/mL after 24 h.

### 3.7. Senescence

The presence of an inflammatory environment could affect normal cell growth, inducing the cells to enter in a state of irreversible arrested growth with altered functions. The senescence activity of cells following their in vitro culture was evaluated by the β-galactosidase staining at selected time points of 3 and 7 days), on PCL-based surfaces. As reported in Figure 8 we observed an in vitro age-dependent increase in SA-b GAL activity. In particular, our results showed that cells cultured on PCL scaffolds presented a lower SA-b GAL activity value compared to the cells cultured on the control ones.

### 3.8. Reactive Oxygen Species (ROS) Production

Analysis of intracellular redox equilibrium was performed to test if PCL surface could induce oxidative stress, able to generate some modifications in signaling pathways. As reported in Figure 9 a weak time-dependent increase in metabolic activity was observed in cells seeded onto both surfaces. During culturing time, when endothelial cells and fibroblasts were seeded onto the PCL-based scaffolds, a lower ROS level was recorded with respect to the value obtained from cells cultured onto control surfaces. Under inflammatory conditions, ROS production increased but, also in this case, the presence of PCL induced a small reduction in the measured ROS value.

### 3.9. Gene Expression

In order to define the molecular signaling underlying the proliferation and regenerative potential of endothelial cells and fibroblasts, the expression of 84 key wound healing associated genes was probed by real-time PCR array, Figure 10 indicated that several genes were differentially expressed in PCL surface compare plastic cultures surface by 2-fold or greater. Among the genes involved in extracellular matrix production collagen type 1, 3, 4, 5, 14 and vitronectin (VTN) were downregulated, whereas metalloproteinase (MMP) were not detected as well as inflammatory cytokines. All the growth factors related to proliferative activity as well as the intracellular pathway were normally regulated in both cell systems and then not reported.

## 4. Discussion

It is urgently needed to develop clinically approved vascular prostheses as alternatives due to the morbidity and mortality caused by vascular diseases and disorders. Nowadays, some commercial artificial blood vessels such as Dacron or e-PTFE grafts have been commonly used for vascular repair [44]. However, the artificial grafts occurred failure for long-term patency and especially for the small-diameter vascular application, because the small-diameter vascular graft increased the danger of thrombosis and occlusion [45]. In the recent years, tissue-engineered scaffolds based on nanofibers have been developed and employed in different biomedical applications [46]. Based on the strategies of tissue engineering, large amounts of tissue-engineered vascular scaffolds with good biocompatibility, controllable mechanical properties, and manageable biodegradability have been designed [47]. Tissue-engineered vascular grafts can be easily manufactured because of the stability and to biomimetic the structure and function of native blood vessels, designing multi-layered vascular scaffolds is an effective way. Amongst these, electrospun scaffolds (ESs) are well known to be one of the most promising strategies to fabricate substrates which resemble native vascular tissues in their geometric alignment. This allows vascular cells to grow in proper orientation [48]. This technique allows for not only the generation of topological fiber arrangement, which mimics the natural extracellular matrix (ECM), but also fabrication of size controllable macrocosmic fibrous grafts [49]. Although the ESs are a reasonably versatile material and highly controlled technology, there are several limitations in the control of morphology for this technique that often determine including poor vaso-activity and undesirable biomechanical properties for handling blood pressure. In the vascular engineering field, it is desired for a hybrid vascular graft substrate to have appropriately sized blood vessels, strong mechanical properties, and bio-activity. To address these problems, a bio-tubular scaffold has been developed by combining an ES system to enhance the mechanical properties of scaffolds generated by printing polymer strands on the ESs substrate [49]. Accordingly, we proposed the fabrication of bilayer vascular grafts by the overlap of two different fibrous layers with different topological and biochemical cues, able to particularize the interface with different population of cells such as endothelial or fibroblasts that coexist in the vascular tissue.

Our preliminary in vitro studies with PCL/Gel fibers confirm angiogenesis of adipose derived mesenchymal stem cells (ADSC) to the gel side. Cells are able to acquire a well-defined mature endothelial phenotype expressing CD31 and von Willebrand factor (detected by means of gene expression and immunohistochemical analyses), are able to organize in a monolayer structure as detected with SEM and to produce basal lamina and growth factor such as VEGF as confirmed by means of gene expression and ELISA. Otherwise the outside layer seeded with fibroblast confirmed not only the good biocompatibility but also the ability to support the production and organization of a good ECM as gene expression and immunohistochemistry for the mail ECM component (collagen fibers) confirmed.

## 5. Conclusions

In this work, we have investigated the in vitro response of endothelial cells and fibroblasts to validate the use of bilayer electrospun scaffolds as vascular graft for tissue engineering. We have verified that outer PCL micrometric fibers show an optimal in vitro response of fibroblast in terms of biological recognition and cell infiltration and reduced inflammation. The addition of bioactive signals—mediated by the presence of gelatin into the inner fiber layer—significantly contributes to the in vitro activity of endothelial cells in terms of gene expression and VEGF secretion, overcoming the major limitations of conventionally used scaffolds to support and guide the newly formed ECM formation. All the preliminary studies indicate a promising potential as micro/nanostructured graft for tissue engineering and future in vivo applications.

## Figures and Tables

**Figure 1 nanomaterials-11-00751-f001:**
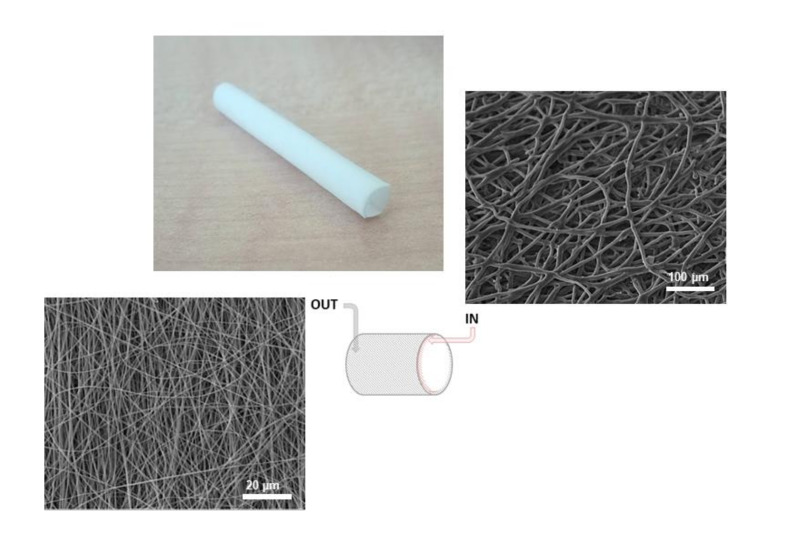
Bilayered electrospun vascular grafts: an overall view of the vascular device and SEM images of electrospun fibers from the inner (i.e., Poly ε-caprolactone (PCL)/gelatin fibers) and the outer (i.e., PCL fibers) layer.

**Figure 2 nanomaterials-11-00751-f002:**
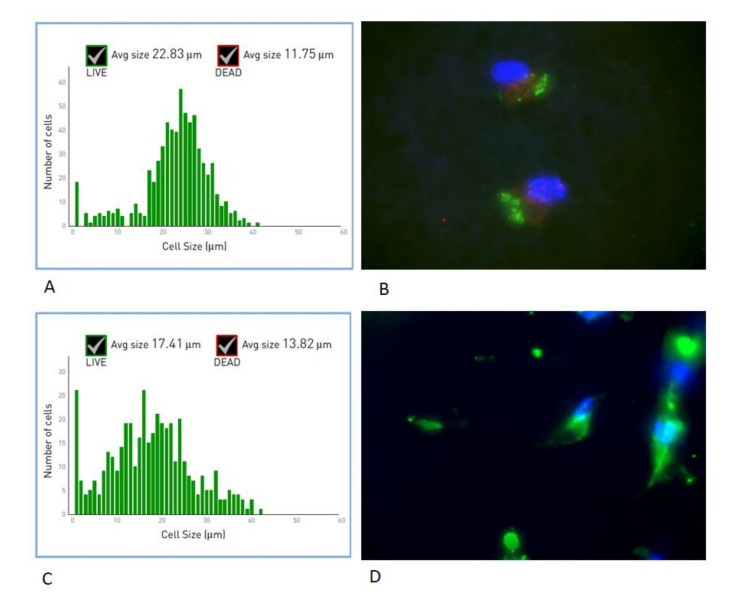
Morphological analyses of endothelial cells (**A**,**B**) and fibroblast (**C**,**D**) after seeding on the electrospun fibers. (**A**,**C**) are related to the size of the cells, (**B**) related to CD31 (green) and wV (red) markers of mature endothelial cells; (**D**) related to phosphatase (green) marker of fibroblast.

**Figure 3 nanomaterials-11-00751-f003:**
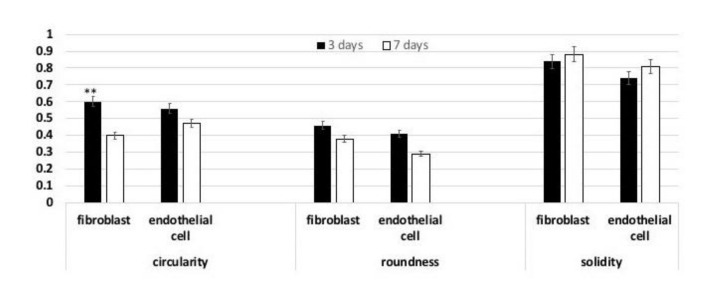
Analysis of the Circularity; Roundness; and Solidity cell shape parameters after 7 and 14 days of culture onto the control and VEGF-enriched implants. ** *p* < 0.05.

**Figure 4 nanomaterials-11-00751-f004:**
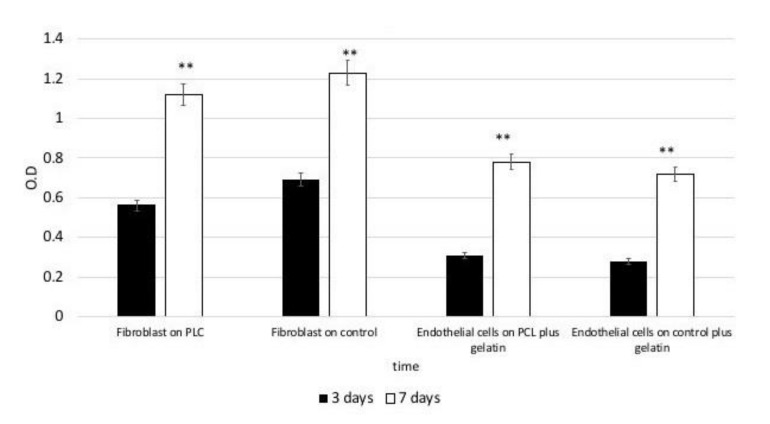
(**A**,**B**) 3-(4,5-Dimethylthiazol-2-yl)-2,5-Diphenyltetrazolium Bromide (MTT) assay. After 3 and 7 days of culture. Cells (Fibroblast and endothelial cells) are able to proliferate on both the PCL surface (fibroblast) or PCL surfaces plus gelatin (endothelial cells) from 3 to 7 days of culture with no statistical difference between the control represented by plastic cultures surfaces or plastic culture surface plus gelatin. Statistically significant differences are indicated as ** *p* < 0.01, and compared with the control condition.

**Figure 5 nanomaterials-11-00751-f005:**
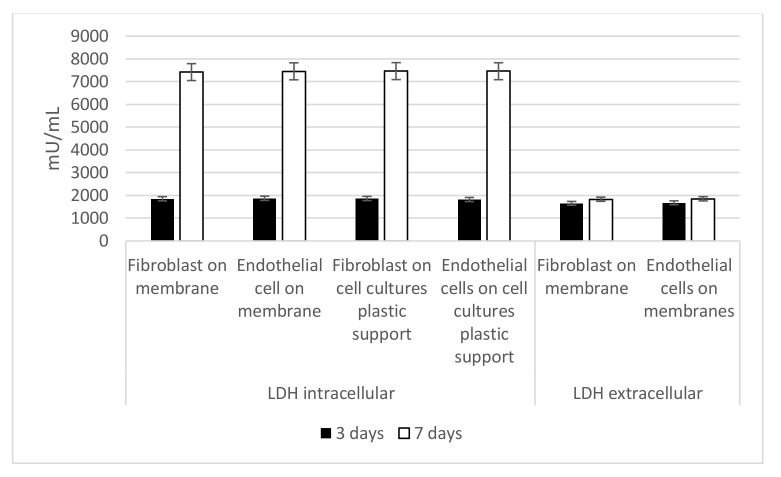
Quantification of intracellular and extracellular Lactate Dehydrogenase (LDH) activity. Intracellular LDH activity proves that cells are able to produce metabolites if seeded onto both surfaces, with improved results after 7 days from seeding. Extracellular LDH activity confirms that metabolites are secreted by the cells and are not associated with damage of the membrane.

**Figure 6 nanomaterials-11-00751-f006:**
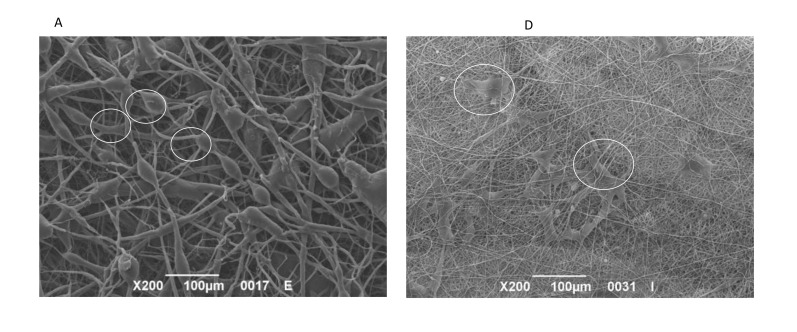
SEM images of Endothelial cells (**A**–**C**) and Fibroblast (**D**–**F**) after 3 days of culture on PCL plus gelatin at 200× (**A**,**D**), 2000× (**B**,**E**) and 5000× (**C**,**F**).

**Figure 7 nanomaterials-11-00751-f007:**
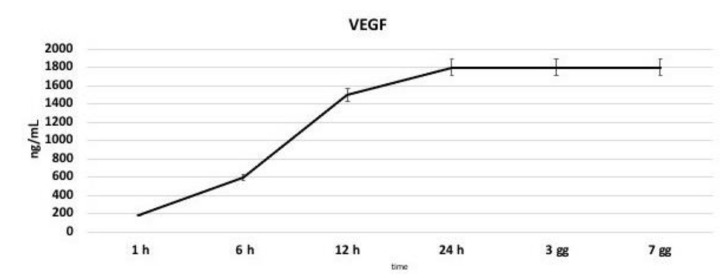
Quantification of Vascular Endothelial Growth Factor (VEGF) release in the cell culture medium. VEGF concentration increases in the medium in a time-dependent manner, reaching a plateau after 24 h.

**Figure 8 nanomaterials-11-00751-f008:**
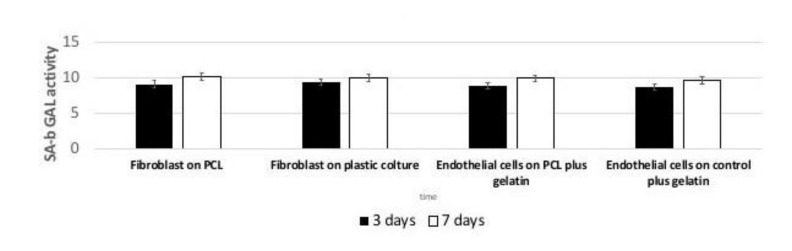
Evaluation of Senescence-associated beta-galactosidase (SA-b GAL) activity of Fibroblast or endothelial cells. Results show that cells cultured on PCL, PCL plus gelatin, plastic cultures, plastic cultures plus gelatin have similar SA-b GAL activity value.

**Figure 9 nanomaterials-11-00751-f009:**
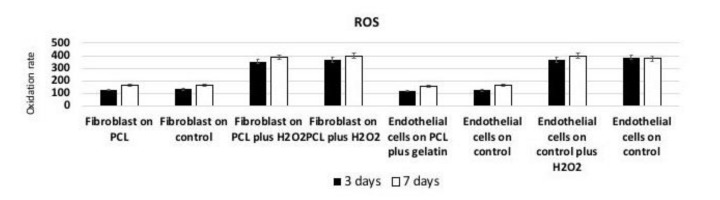
Reactive Oxygen Species (ROS) production of Fibroblast and endothelial cells seeded onto PCL, plastic cultures, PCL plus gelatin, plasticultures plus gelatin. Histograms show a slight time-dependent increase in metabolic activity in cells seeded onto both surfaces. Results are expressed as fluorescent arbitrary units per second confirm no production of pathological concentration of ROS.

**Figure 10 nanomaterials-11-00751-f010:**
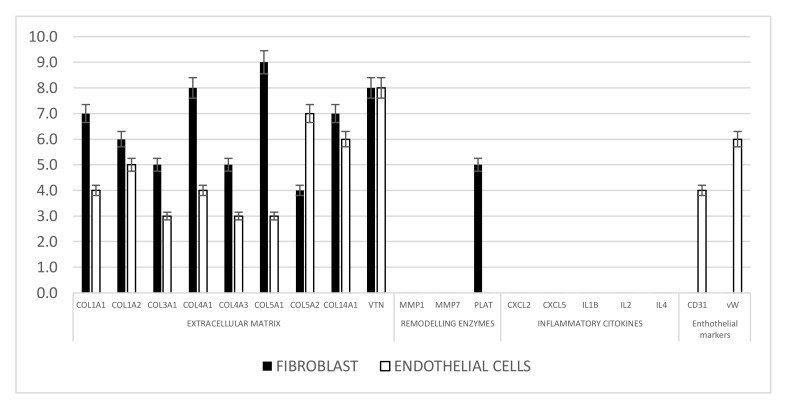
Real-time PCR analysis of extracellular matrix and remodeling enzymes markers in Fibroblast and endothelial cells cultured on PCL electrospun surfaces. Results for each experiment are obtained from triplicate experiments and values are expressed as the mean ± SD.

**Table 1 nanomaterials-11-00751-t001:** Summary of processing parameters used for the fabrication of flat membranes from inner (IN) and outer (OUT) graft layers.

Layer	Flow Rate (mL/h)	Voltage (kV)	Electrode Gap (mm)
IN	0.1	13	130
OUT	0.5	15	150

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
