# Peer review of "Electrospun PCL-Based Vascular Grafts: In Vitro Tests"

_nanomaterials, 2021, doi:10.3390/nano11030751_

Round 1

Reviewer 1 Report

In this manuscript, the authors reported the fabrication of PCL and PCL/Gelatin fibrous bilayered tubular scaffolds by electrospinning and investigated its application for vascular tissue engineering. The in vitro test results seemed interesting and meaningful. Before accept for publication, some concerns should be clarified.

1.      The real image of the prepared tubular should be provided in Figure 1.

2.      The electrospinning process should be revised to tell how to prepare the tubular scaffolds. And Line 100 in page 3, “Process parameters used were summarized in Table 1.” The mentioned Table 1 was missing.

3.      Please clarify the in vitro test based the electrospun meshes or the tubular scaffolds?

Author Response

  1. The real image of the prepared tubular should be provided in Figure 1. 

Thank you for the comment. A picture of the electrospun graft was included in Figure 1.

  1. The electrospinning process should be revised to tell how to prepare the tubular scaffolds. And Line 100 in page 3, “Process parameters used were summarized in Table 1.” The mentioned Table 1 was missing.

Thank you for the comment. The preparation procedure has been revised and the table (only cited) is onw included into the section 2.2 of Materials and Methods.

3.      Please clarify the in vitro test based the electrospun meshes or the tubular scaffolds? 

Thank you for the comment. The the in vitro test based the electro spun meshes

Reviewer 2 Report

The manuscript by Barbara et al. described the fabrication of PCL and PCL/gelatin fibers for the potential uses of vascular grafts. The authors performed fiber productions and basic materials characterizations with the majority of the work in cell studies. The manuscript has some drawbacks, and the authors must address these issues before being considered for publication.

  • In section 2, Materials and Methods, the authors need to start with a subsection of 2.1 Materials. There were no information on where the authors obtained the materials and supplies for the study.
  • Based on the information in the current section of 2.1. Subsection PCL scaffolds production, it is very difficult to understand why the authors microsized fibers at the outer layer and sub-micron sized fibers at the inner layer as shown in section 3.1. What makes the difference in fiber diameter? Please describe the graft making process in detail.
  • Also in section 3.1, the authors mentioned “Moreover, the addition of Gelatin to the PCL solution allows to evidently reduce instability phenomena of the polymer jet during the deposition process, thus guaranteeing the formation of bead less fibres without relevant surface irregularities”. This is not the case here since PCL, as a linear polymer, is fairly easy to electrospin (again, not sure what type of PCL the authors used). Gelatin, a non-linear polymer with a rigid backbone, is often hard to electrospin. Why would adding gelatin to PCL allow less beading? It should be the opposite.
  • Figure 4: Can the authors rearrange the control groups to the very left of the graph? This will make the graph more understandable.
  • Figure 5: Should there be a control group?
  • Section 3.6: If the SEM observation was at 1 day on cell attachment, please do not elaborate on “…exhibiting a good ability to migrate and to secrete exosomes (black circles) onto their surfaces.”
  • Section 3.6: Both endothelial and fibroblast cells are in the 10 – 20 um range. Figure 6 did not show the corresponding images of the cell attachment. Figure 6A is fiber beading (contrary to authors’ statement in section 3.1). Features in Figure 6C and 6F were in nanoscale, definitely not cells.

Author Response

  1. In section 2, Materials and Methods, the authors need to start with a subsection of 2.1 Materials. There were no information on where the authors obtained the materials and supplies for the study.

Thank you for this comment. A section 2.1 was included where all the materials used were listed.

  1. Based on the information in the current section of 2.1. Subsection PCL scaffolds production, it is very difficult to understand why the authors microsized fibers at the outer layer and sub-micron sized fibers at the inner layer as shown in section 3.1. What makes the difference in fiber diameter? Please describe the graft making process in detail.

Thank you for this comment. the section 2.2 has been amended improving the description of the electrospun graft preparation as requested. Difference in fibre diameter is just reported in the section 3.1 of the manuscript:“Quantitative measurements via image analysis reveal an evident difference in the fibre diameters from micrometric (5.11 ± 0.47) µm to sub-micrometric size scale (0.822 ± 0.11) µm, ascribable to the effect of used solvents with different dielectric permittivity – i-e., chloroform or HFIP”. A new sentence has been included in order to clarify what makes the difference in fibre diameter: “In particular, higher permittivity of HFIP allows to generate stronger interactions among polymer chains under the applied electrical forces, so promoting an higher stretching of the polymer jet during the deposition, and ultimately, a reduction of the final diameter of fibres as reported in previous works [40]”

  1. Also in section 3.1, the authors mentioned “Moreover, the addition of Gelatin to the PCL solution allows to evidently reduce instability phenomena of the polymer jet during the deposition process, thus guaranteeing the formation of bead less fibres without relevant surface irregularities”. This is not the case here since PCL, as a linear polymer, is fairly easy to electrospin (again, not sure what type of PCL the authors used). Gelatin, a non-linear polymer with a rigid backbone, is often hard to electrospin. Why would adding gelatin to PCL allow less beading? It should be the opposite.

Thank you for your comment. As the reviewer underlined, polymer chain linearity as well as molecular weight distribution can influence the morphology of fibres. In particular, linear polymers such as PCL and strict molecular weight distribution are more easily to process, reducing the occurrence of instability phenomena and the formation of beads along fibres. However, this effect is strongly influenced by the solvent properties. In this case, the use of high permittive solvents allows to reduce the fibre diameters of PCL, also promoting the formation of beads along fibres due to presence of significant whipping instabilities (see ref. 40). In this case, the presence of high polar groups along the gelatin backbone allows to influence the local interactions among solvent molecules and PCL chains, so reducing the formation of beads along fibres under the applied electrical forces. Accordingly, the sentence in the section 3.1 has been revised and supported by three new appropriate references (refs 41- 43) as follows: “Moreover, the combination of Gelatin macromolecules with PCL in solution allows to drastically reduce instability phenomena of the polymer jet, typically induced by the use of highly permittive solvents [41], thus guaranteeing the formation of defect less fibers with more homogeneous fiber diameter distributions, in agreement with experimental evidences on similar bicomponent fibres [42 , 43]”.

  • Figure 4: Can the authors rearrange the control groups to the very left of the graph? This will make the graph more understandable.

Thank you for your comment. We done it

  • Figure 5: Should there be a control group?

Thank you for your comment. We done it

  • Section 3.6: If the SEM observation was at 1 day on cell attachment, please do not elaborate on “…exhibiting a good ability to migrate and to secrete exosomes (black circles) onto their surfaces.”

Thank you for your comment. We done it

  • Section 3.6: Both endothelial and fibroblast cells are in the 10 – 20 um range. Figure 6 did not show the corresponding images of the cell attachment. Figure 6A is fiber beading (contrary to authors’ statement in section 3.1). Features in Figure 6C and 6F were in nanoscale, definitely not cells.

Thank you for your comment. We images are related to cells, we add the bigger SEM images 

Reviewer 3 Report

Nanomaterials - 1094166

Electrospun PCL based vascular grafts: preliminary in vitro test

In this study authors study in vitro how endothelial cells and fibroblasts grow into electrospun PCL and PCL/Gelatin scaffolds. Scaffolds with various fiber size were produced and used for cell culture. Cell morphology and proliferation was then assessed at 3 and 7 days.

General comments

The paper presents some interesting data. Producing a fibrous durable vascular graft of small diameter remains a huge medical challenge. Electrospun constructions (replace electrospinned with electrospun in the title) are interesting potential candidates within that research strategy. However several concerns and questions remain regarding the manuscript.

Introduction

-A thorough state of the art about the trials using electrospun scaffolds for vascular applications and published in literature is missing. This would help defining properly the required specifications for the design of an optimal ES graft: pore size, fiber size, surface density, strength.

-Line 85: “to to…”

Materials and Methods

-Line 92: what is the need, micro? macro ? a mix ? which size provides which property ?

-not clear what was tested with cells: membranes ? tubes ?

-which cells were used ? which animal ? how long was the culture (appears only in results, should be defined in MM section)

-why using a hybrid PCL/ Gelatin on the inner side and a sole PCL on the outer side ? all this should be documented.

-what’s the purpose of using larger fibers outside, and smaller fibers inside ?

-Line 153: “…in bone sample…” what has bone to do with his study….?

-Line 165: replace “whose” by “which”

Results

-Line 228: what is the Gelatin proportion ? and this is Materials and Methods …

-not clear if the Gelatin is combined with PCL in the fiber production or if gelatin is coated on the scaffold surface…this should be clarified and justified

-Lines 235-236: again, explanation should be given about why fibroblasts on the PCL scaffold and endothelial cells on the hybrid scaffold

-Lines 239-240: replace nano with micro

-Lines 251-254: the seeding is performed on membranes ? tubes ?

Discussion

-The discussion remains superficial and should be deepened and compare in more details the results obtained here with other existing works. Moreover, further scientific explanations regarding the cell ingrowth pattern based on the fiber morphology should be given.

Author Response

Introduction

-A thorough state of the art about the trials using electrospun scaffolds for vascular applications and published in literature is missing. This would help defining properly the required specifications for the design of an optimal ES graft: pore size, fiber size, surface density, strength.

thanks, done

-Line 85: “to to…”

Thanks, done

Materials and Methods

-Line 92: what is the need, micro? macro ? a mix ? which size provides which property ?

-Line 92: what is the need, micro? macro ? a mix ? which size provides which property ?

Thank you for this comment. In this work, it is proposed a bilayered tube to offer optimal materials interfaces with characteristic fibre/mesh sizes to support the biological activities of endothelial and fibroblast cells for the in vitro regeneration of vascular tissue. This has been clarified in the discussion by a new sentence as follows: “Accordingly, we proposed the fabrication of bilayer vascular grafts by the overlap of two different fibrous layers with different topological and biochemical cues, able to particularize the interface with different population of cells such as endothelial or fibroblasts that coexist in the vascular tissue

Moreover, this is further specified in the conclusion section as follows: “In this work, we have investigated the in vitro response of endothelial cells and fibroblasts to validate the use of bilayer electrospun scaffolds as vascular graft for tissue engineering. We have verified that outer PCL micrometric fibres show an optimal in vitro response of fibroblast in terms of biological recognition and cell infiltration and reduced inflammation. The addition of bioactive signals - mediated by the presence of Gelatin into the inner fibre layer – significantly contributes to the in vitro activity of endothelial cells in terms of gene expression and VEGF secretion, overcoming the major limitations of conventionally used scaffolds to support and guide the newly formed ECM formation. All the preliminary studies indicate a promising potential as micro/nanostructured graft for tissue engineering and future in vivo applications”.

-not clear what was tested with cells: membranes ? tubes ?

membranes

-which cells were used ? which animal ? how long was the culture (appears only in results, should be defined in MM section)

Thanks, we added a new section on MM

-why using a hybrid PCL/ Gelatin on the inner side and a sole PCL on the outer side ? all this should be documented.

Because endothelial cells need gelatin substrates to attached. So since we want to reproduce a membrane with the same cellular structures of native vessel where endothelial cells are in the inner side and the fibroblasts in the outer side, we treated only a side with gelatin

-what’s the purpose of using larger fibers outside, and smaller fibers inside ?

the fibroblast that will be in the outer side to produce collagene type I need larger fibers, by contrast, endothelial cells that need to form the endothelial wall need of smaller fibers

-Line 153: “…in bone sample…” what has bone to do with his study….?

It is a refuse sorry

-Line 165: replace “whose” by “which”

thanks, done.

Results

-Line 228: what is the Gelatin proportion ? and this is Materials and Methods …

Thank you for this comment. The relative PCL/Gelatin ratio has been included in the section 2.2 (Materials and Methods)

-not clear if the Gelatin is combined with PCL in the fiber production or if gelatin is coated on the scaffold surface…this should be clarified and justified

Thank you for the question. Gelatin is dissolved in solution with PCL to form blended fibres. This has been clarified either in the description of the processing methods (section 2.2) and in the results (Section 3.1).

-Lines 235-236: again, explanation should be given about why fibroblasts on the PCL scaffold and endothelial cells on the hybrid scaffold

Thanks, done

-Lines 239-240: replace nano with micro

Thanks, done

-Lines 251-254: the seeding is performed on membranes ? tubes ?

Membranes

-The discussion remains superficial and should be deepened and compare in more details the results obtained here with other existing works. Moreover, further scientific explanations regarding the cell ingrowth pattern based on the fiber morphology should be given.

Thanks, done

Reviewer 4 Report

The authors evaluated electropsun PCL and PCL/gelatin tubular scaffolds as a vascular grafts. PCL has been used to fabricate vascular grafts (see Circulation 2008 118, 2563-). Works on electrospun PCL or their blend with collagen/gelatin are also available in the literature. Two-layers structure is not unusual as well. The novelty of this work is fairly low from the perspective of biomaterials. The authors used PCR to demonstrated ECs and FBs did not dedifferentiation. The contribution of this portion does not significant for the publication of this work. The results are fairly obvious given the knowledge provided by many articles on using electrospun PCL as vascular grafts. 

Author Response

We processed Poly ε-caprolactone (PCL) via electrospinning for the production of bilayered tubular scaffolds for vascular tissue engineering application. Endothelial cells and fibroblasts were seeded into the two side of the scaffolds: endothelial cells onto the inner side composed of PCL/Gelatin fibres able to mimic the inner surface of the vessels, and fibroblasts onto the outer side only exposing PCL fibres. Extracellular matrix production and organization has been performed by means of classical immunofluorescence against collagen type I fibers, Scanning Electron-Microscopy (SEM) has been performed in order to evaluated ultrastructural morphology, gene expression by means real time PCR has been performed to evaluate the phenotype of endothelial cells and fibroblasts. Results confirmed that both cells population are able to conserve their phenotype colonizing the surface based on electrospun PCL fibres supporting the hypothesis that bilayered scaffolds should be a good candidate for vascular surgery.

The novelty of our article is that up to now no bilayered scaffolds based on electrospinned technology has been never tested with two different type of cells.

We add several new parts.

Round 2

Reviewer 2 Report

The authors have addressed the issues suggested/recommended, and therefore, there is no more questions from the reviewer. Good work!!

Reviewer 4 Report

No comments.